# Between Empowerment and Gentrification: A Case Study of Community-Based Tourist Program in Suichang County, China

**Zijing Zhao [1], Yan Wang [2], Yuxian Ou [2] and Lucen Liu [2,\*]**

[1] School of International Studies, Zhejiang University, Hangzhou 310058, China; tinazhao0328@hotmail.com
[2] College of Education, Zhejiang University, Hangzhou 310058, China; wy19940903@gmail.com (Y.W.); eveqiya@163.com (Y.O.)
\* Correspondence: lucenliu@icloud.com

**Abstract:** The phenomenon of hollow villages is a long-lasting obstacle to China's rural development. With this background, this study examines a for-profit community-based tourist program operated at a rural hollow village in Zhejiang, China and explores how this program facilitates meaningful transformations in the community. The theoretical concept of empowerment was introduced to critically understand and analyse the community transformations, and the data was collected through program-related or village-related media content, participant observation, and focus group interviews. Our findings reveal that varied types of empowerment had been gained by the residents via the program, which has strongly demonstrated the positive and meaningful transformations in the community. Lastly, gentrification, a type of community transformation, was a positive change from the program managerial staff's view, but it could lead to uncertainties and problems of the economic and political disempowerment to the residents in the long term.

**Keywords:** China; rural community; tourism; empowerment; transformation; gentrification

## 1. Introduction

When the People's Republic of China was founded in 1949, it was largely a rural country, with farmers and villages as its foundation [1]. Still, rural development and rural poverty alleviation have been key tasks for the country's policymakers [2]. A long-lasting obstacle to rural development is the phenomena of hollow villages, which emerged in the early 1990s, and created economic, societal, and environmental problems [3,4]. This phenomenon resulted from the industrialization and urbanization, which drove a large number of young rural workers to migrate to the urban areas and left the elders and children in the villages [5]. It wasted agricultural land, endangered rural life, widened the economic and social gap between the urban and the rural, and it might lead to social instability [6]. Hence, over the years, the Chinese government issued a series of policy documents to support the economic and social development in the rural areas and to increase the human and capital mobility between the urban and rural areas [7,8].

Community-based tourism has emerged from the community development strategy, aiming to use tourism as a tool to manage tourism resources with the participation of the local residents, and at the same time, improve the life of local community [9]. Using a small-scale case study to explore a rural community-based tourist program, this paper examines what changes the program has made to the community and how these changes have empowered or disempowered the residents. On this basis, we discuss how this program may facilitate meaningful transformations in the community.

We chose this newly-opened tourism program, 'Mount Banner and the Hermit Master', as the case because it is based at an underdeveloped village, or a hollow village called the Tea Village. The village is located in Suichang County, whose GDP ranked 81st among the 90 counties in Zhejiang Province in 2021 [10]. This program is designed by a tourism

investment and management company, which has planned to develop Tea Village into a rural tourism destination featured with varied activities, including traditional house experiences, recreational agricultural activities, martial arts practices, organic food tasting and national park hiking. This program has two aims: to provide tourists with the traditional rural lifestyle, and to achieve sustainable development in the village [11]. Inspired by the second aim of this program, this study explores to what extent the community-based tourist program may cause meaningful rural development.

Tourism has the potential to meaningfully, positively, and sustainably transform communities, yet without empowering the communities and their residents, tourism development may not be regarded as meaningful and beneficial for the locals [12]. Hence, we argue that to critically understand the community transformation induced by business practices of a tourist program, the concept of empowerment is needed. This study also contributes to the limited understanding of "how empowerment occurs among individuals and communities" and "what makes residents feel empowered through tourism" [13]. Further, this study identifies the uncertain transformation in the community resulting from tourism gentrification and local residents' self-gentrification [14,15].

In what follows, we review the scholarship on tourism for development and the impacts on local communities and then introduce the theoretical concept of empowerment and the research methods. Findings are divided into two parts, including changes in the community at the program's preparation stage and at its operational stage. Lastly, we critically discuss how these changes have empowered residents and how gentrification in the community may be considered as an effect of disempowerment to the residents in the long term.

## 2. Tourism for Development and the Impacts on Communities

During the last decades, local governments around the world have adopted tourism as a development drive. The practice of community-based tourism has been recognised as useful in promoting the community's all-round development [16], even though there was a mixture of positive and negative impacts on the communities and residents.

Positive impacts of community-based tourism include poverty alleviation [17,18], improvement of social justice [19], and post-disaster resilience building [20]. Scholars remind that economic benefits alone, such as increased incomes and employment, are not enough to evaluate the positive impact of tourism development on the community [21]. The evaluation of tourism benefits also needs to include the residents' knowledge, education, skills and awareness [22,23].

Community-based tourism may also create negative impacts on the communities' environment, economy and social life, such as deteriorated natural environment, low-paying jobs, traffic problems, increased living costs, crowdedness, and conflicts between residents and tourists [24,25]. These problems could be inter-related and cause profound influences. For instance, tourism could transform the traditional livelihood from fishing-and-agricultural-dependent economies to economies overtly dependent on the service industries [26,27]. Consequently, the traditional knowledge and skills attached to the livelihood became unsustainable [28]. When external and noncontrollable events occurred, such as the COVID-19 pandemic, the tourism-dependent communities could become rather vulnerable.

The negative impacts and unsuccessful implementation of community-based tourism development could be caused by unequal power relations, asymmetry of knowledge and information, and unmatched interests between tourism investors and communities [29,30]. The imbalanced power relation among government, external tourism developers and communities has been deemed a fundamental problem [31], considering that communities often had low-levels of participation in planning and decision-making stages and consequently benefited less from its tourism development [30,32–34]. To enable the successful and sustainable implementation of community-based tourism, growing research has em-

phasised the importance of establishing partnerships with communities and empowering the residents collectively [10,35,36].

Research on community-based tourism in China also found that governments and companies often impose or single-handedly decided the tourism development [29,30,37]. Community-based tourism in China often generated a phenomenon of socio-cultural transformation—that being gentrification [14,38–40]. Gentrification originally describes that the wealthier middle-class people or gentry move to and invest in the working-class neighbourhood, and as a result, it changes the landscape, demography, economics and cultural characteristics of the neighbourhoods [41,42]. Research on gentrification of urban residents mostly demonstrated negative impacts, such as residential relocation, cultural dimension of displacement, social polarisation, rising rent, disruption of residents' everyday lives and residential population decline [43,44]. In contrast, research on tourism gentrification in China mainly focused on the rural and ethnic minorities' areas and the merits of self-gentrification among residents. Such self-gentrification includes improving residents' socio-economic status, maintaining residential population and their culture, and enhancing collaboration between locals and newcomers [14,38]. In this sense, self-gentrification could be seen as a form of self-empowerment. Nevertheless, we question this uncritical view on gentrification in the rural areas in our later findings.

## 3. The Conceptualisation of Empowerment Framework

Empowerment can be generally defined as individuals' or groups' capacity and competency to determine their own affairs and influence their surroundings [45,46]. Most research on tourism empowerment in the Chinese context has employed Scheyvens' [47] four-dimensional tourism empowerment framework, which includes economic, psychological, social and political dimensions (e.g., [48,49]). This study proposes to conceptualise and use a more comprehensive empowerment framework that adopts different forms, levels and sources of empowerment (see Table 1). The different forms of empowerment include psychological, behavioural, economic, political, environmental and knowledge empowerment [45–47,50,51]. These forms of empowerment can be categorised into different levels, such as individual, collective, organizational/institutional, structural, family or community levels [35,52,53]. Because there are overlapping meanings among these levels, this study adopts the pair of individual and community levels.

**Table 1.** The conceptualisation of empowerment framework.

| Levels of Empowerment | Forms of Empowerment | Source of Empowerment |
| --- | --- | --- |
| Individual level | Psychological, economic, knowledge forms | From the outside and/or inside |
| Community level | Social, political and ecological forms | |

At the individual level, there are mainly psychological, economic and knowledge forms of empowerment. Psychological empowerment enhances self-confidence and satisfaction, self-esteem, dignity and pride [13,52]. For example, tourists' recognition and appreciation of the communities' natural environment, culture or history would empower the residents psychologically via their heightened pride and self-esteem. Economic empowerment refers to monetary gains through employment or work opportunities created by tourism [47]. For developing countries and areas, economic empowerment from community-based tourism is the communities' main concern [47,54]. Economic empowerment also happens at the community level, but scholars suggest that it is important to consider economic empowerment at the individual or family level because the already marginalised and poor individuals or families may not equally share the financial benefits from tourism development [55,56]. Knowledge empowerment is about receiving education, obtaining new information and increasing awareness of opportunities [32,57].

Equipped with proper knowledge, residents have a higher willingness to be involved in the decision-making and planning stage of tourism development [58].

At the community level, there are mainly social, political and environmental forms of empowerment. Social empowerment occurs when "a community's sense of cohesion and integrity has been confirmed or strengthened by an activity", such as tourism and tourism-related infrastructure projects [47] (p. 248). However, tourism may disempower the community by producing social disharmony, such as crime, begging, crowdedness, as well as the loss of local culture and traditions [47]. Political empowerment happens when the community collectively shifts the unbalance power between the investors and residents by influencing the tourism development process [10,47]. Due to the close link between social and political empowerment, Rocha [51] conceptualises the socio-political empowerment to be something that "focuses on the process of change within a community locus in the context of collaborative struggle to alter social, political, or economic relations" (p. 37). Lastly, environmental empowerment refers to tourism's contribution to the sustainability of the communities' ecology [59].

Regarding the relation between individual and collective levels of empowerment, the former serves as the foundation of the latter [51]. However, individual and collective empowerment may happen at the same time. For example, tourists' recognition of local cultural traditions heightens the confidence and pride at both individual and community levels [45]. In addition, we anticipate that different forms of empowerment could interact with each other within the same level or across levels. For example, knowledge empowerment increases confidence and, consequently, facilitates social–political empowerment, which affects decision-making bodies of tourism development and shifts the power relations.

In addition, empowerment comes from outside and inside sources. In the rural communities, empowerment often firstly comes from the outside. For example, an outside form of empowerment could be generated when the more powerful outside investors or government authorities invited the residents to participate in tourism decision-making process, which could also be the first step of tourism empowerment. Prior study has confirmed that residents' higher participation would lead to stronger economic and psychological empowerment. An inside form of empowerment could be that residents identify the problems faced by the community and seek solutions through taking control of the tourism development. Both outside and inside empowerment can create meaningful transformation in the community.

With this empowerment framework, this study examines which types of empowerment were produced through this community-based tourist program and to what extent the community has been transformed.

## 4. Tea Village and the Tourist Program

Tea Village has a 400-year history and is located at a remote mountain national reserve in Zhejiang Province, China (see Figure 1), where public transportation is not convenient. In 2016, the population registered in Tea Village was around 140, but the regular resident population was only about 50, and most of them were elderly people. Almost all the children left with their parents to nearby larger towns. A household normally had two to three houses and/or huts, which were used for living, cooking or storing, respectively. Due to the outflow of labour to neighbouring towns and cities, many houses stood empty.

In 2016, an owner of a tourism and hospitality company visited the village, and in the same year, the company decided to develop a community-based tourist program in the village. Before that, the company had developed varied tourism and leisure programs at towns and tourism areas around China. The Tea Village's relatively intact traditional housing, well-preserved surrounding, natural environment and martial arts tradition were regarded as attractions for a tourist program. Based on the village's martial arts tradition and surrounding mountains, the tourist program is named 'Mount Banner and the Hermit Master', and it targets urban tourists. The program claims to be the first one based at a

hollow village in China and a resident-centred tourist program (see Table 2 for the actions of program implementation). Although the tourism is based at the village, the village itself is free and open for people to visit. Nevertheless, the program adopted a membership system, meaning that one needs to be a paid member if he or she wishes to stay at the houses and participate in the activities run by the program.

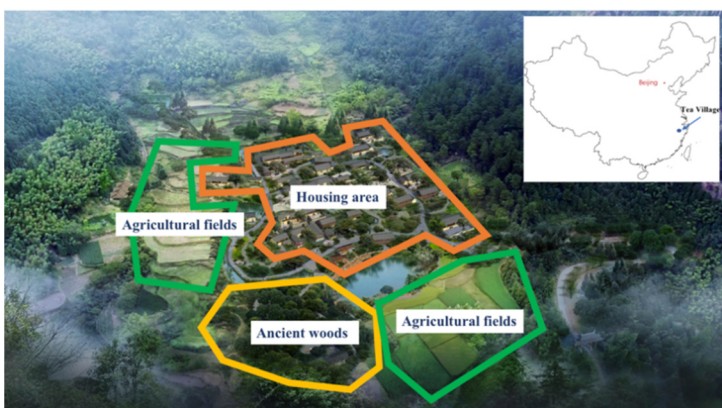

**Figure 1.** The location of Tea Village and the planned areas of the village-based tourist program. (Figure was drawn according to the map and on-the-spot investigation).

**Table 2.** The actions of the program implementation at different stages in the village.

| Stages | Actions of the Program Implementation |
| --- | --- |
| 2016 to 2019 | The program negotiated the house rent, land lease with residents, renovated residents' houses, built new houses, and set up a residential committee. |
| 2019 to present | The program introduced activities, such as traditional house experience, recreational agriculture and the martial arts experience. |

## 5. Methods

This study has adopted a qualitative approach to gain in-depth understandings of the changes that the program brings to the community. With the ethics approval from the researchers' university, secondary and primary evidence was collected between February 2021 and November 2021. Secondary data was collected from printed and online materials about the village, the tourism and hospitality company and the program, the forms of which include books, news, articles tourist guidebooks and conceptual planning.

After familiarising themselves with the secondary data and generating the observation and interview guides (see Appendix A), researchers started primary data collection through fieldwork and focus group interviews. All researchers conducted two days of fieldwork to gain first-hand experience of the village and the program, as well as to make acquaintance with potential interviewees. In the field, we mainly used participant observation and our roles shifted between complete participants, participant observer and complete observer, depending on our capabilities to conduct the program activities. For instance, we were complete participants while staying at the traditional style guest houses and learning simple martial art moves; we were complete observers of others practising difficult stick-fighting moves and green tea making processes; we were participant observers while visiting martial art museums, agricultural fields and residents' homes. During the participant observation, field notes, photos and videos were taken with the permission of participants.

Two focus groups were conducted during the fieldwork. Focus group interview was an efficient method to elicit discussion among interviewees about their experiences of the village before and after the program, and it allowed researchers to gain deeper understandings of the changes in the individuals and community. We used the notes from participant observation to update the interview guides.

The first focus group invited the company owner (male) and 2 managers (females) and was conducted on day 1 after we experienced a whole day of activities. The second focus group was conducted in the middle of day 2 and invited a nonresident employee (male), three resident employees (a female, two males) and two resident non-employees. All the participants were purposefully invited in view of their experiences of working in the program and/or living in the village. All of them have experienced the preparation and operation stages. Hence, their experiences are helpful for us to understand the situations before and after the program was operated. Most of the elder resident non-employees spoke the local dialect, which the researchers were unable to speak. Hence, it was a limitation that we could not invite more resident non-employees. The only two resident non-employees that participated in the focus group had regular interactions with the company staff in a residential committee. Except for a younger manager and the nonresident employee, the rest of the interviewees are middle-aged (above 40). Two focus groups lasted 60.57 min and 41.08 min, respectively, and were recorded by smart phones with the permission of participants.

Data was analysed following a six-step theory-drive thematic analysis [60,61]. First, we familiarised ourselves with the textual data, including the fieldnotes, self-reflections, and focus group transcripts. Second, we used a deductive way to code the texts and produced the initial empowerment framework-driven codes, such as economic growth, environment protection, self-confidence, etc. Third, the related codes were classified into the different dimensions of empowerment, and fourth, the shared meanings were searched from the grouped codes to form initial themes. Fifth, we repeated the third and fourth steps by reviewing codes and themes. Sixth, three themes were finalised, illustrating the relationship between (dis)empowerment and community transformation, which will be discussed below.

## 6. The Changes of the Community at the Program's Preparation Stage

The initial changes of Tea Village started from the outside drive, namely the external tourism company and its program. There were economic changes (e.g., residents' increased income), physical changes (e.g., the renovation of residents' houses and the establishment of new houses for residents), social changes (e.g., the establishment of residential committee to hear residents' opinions on the program), and population changes (e.g., the returning residents from adjacent towns).

Regarding the economic change, before this program (2016), the per capita income of residents was around 9000 yuan, and in 2019, the per capita income of residents in the labour force reached around 30,000 yuan [62]. At the preparation stage (around the end of the year 2016), the company introduced the ways of economic gains to the residents through the program, including the rental income, land lease income, the selling of agricultural products and employment wage. First, for the rental income, the company signed a twenty-year lease with each house owner, who can receive an annual payment about 15,000 RMB. Five years after the lease signing date, the rent will increase by 10% every three years. After 20 years, villagers can choose to renew the lease or take their houses back. Second, the company rented the unused agricultural field (40,002 m$^2$) from residents for 0.9 RMB per square meter per year and the tea field for 1.8 RMB per square meter per year. Third, the company helped the residents to sell local agricultural products, such as winter bamboo shoots, bamboo barrel aged spirit, green tea, kiwi fruits and honey. Fourth, residents who are available to work could become the program employee and earn a monthly wage about 3000 RMB. Depending on their preference for involvement in the tourist program, residents can acquire different types of income. Fifth, the company purchased the agricultural products from residents—such as tea leaves, bamboo shoots, chickens and eggs—and sells them online to the urban areas by the program staff. Sixth, at the time of this research, 30 residents have become the program employees and six of them are returning residents. Most of the resident employees worked in the areas of agriculture, maintenance, room

service, cleaning, carpentry, masonry, security and other manual work. Sometimes, the program would temporarily hire more residents for short-term projects [63].

Regarding the physical change of the community's appearance, the company owner pointed out that all the families have signed the 20-year house lease (24 in total), except one who has already rebuilt the modern-style concrete-and-brick house before the program's preparation stage. According to the company owner, the advantage of Tea Village lies in its preserved look of traditional Chinese mountainous village. The company owner reflected on the following:

> At the end of 2016, I first came here. ... it's basically a hollow village and some houses looked a little shabby and some walls were collapsed. The advantage of that is it retains the original look of a small Chinese village and this is valuable.

By renting residents' traditional houses, the program planned to retain the original look of the village by minimally repairing the houses' exterior mud walls. However, residents did not understand the program's appreciation and protection of the old houses. The company owner and managers recalled that at the preparation stage, the house owners hoped the company could rebuild their houses with concrete walls and decorate the walls with white tiles. To make compromises, the company renovated the villagers' old houses for the tourists to stay, and it built similar-looking new houses beside their traditional homestead for the owner's family to live in. This contradictory idea of how the houses should be renovated has led to the social change of establishing the residential committee.

To increase residents' involvement, the program supported the establishment of the residential committee, which served as an interface between the community and the company. Before a committee meeting, issues and concerns were gathered from residents or the company. Then, at the meeting, the company representatives discussed the issues with residents. Manager 1 recalled that at the preparational stage, "all the issues would be presented to the residents (at the committee meeting). For example, whether a road was needed to be repaired. This's not a legal problem but need consultation and discussion".

The program also invited academics, lawyers and government officials from outside to the committee meetings to share ideas about rural revitalisation with the community. Resident employee 1 pointed out that for him, "to be able to interact with these elites makes us feel equal. Our improved status is what benefits us the most, not the economic gains." Resident employee 1 also reflected that residents' continued interaction with the urban elites made them broaden their mindset and change behaviours:

> My entire personality has changed. I used to be very fussy about small things, fighting for tiny interests and avoiding responsibilities. Some people changed but some are still like that. If I meet them, I won't fight them for anything or spend too much time dealing things with them.

This narrative of changing mindset and behaviours indicated the increased social harmony. According to the company owner, such changes in the community emerge from the interaction between the urban modern lifestyle and rural traditional lifestyle, which help to develop a modern type of Chinese village. In the later findings, we provide a critical look at the modern transformation through the discussion of gentrification.

Another aspect of the community-level change at this stage was the increasing resident population. Before 2016, the resident population was around 50, and by the end of 2019, the resident population in this village raised to 87. Resident employee 2, a returning resident, said this:

> There is no school nearby. When my son was 6 years old, I took my son to study at the nearby town and we stayed there for more than 20 years. Now, the company came to develop tourism and housing. My husband and I moved back to work for the company. We have our own houses, and we gain both the rent and the wages. Our income suddenly became much higher.

Resident non-employee 2 reflected that before the program, there were very few people staying at the village, and most of them were in their 70s and 80s and the youngest residents were about 50 years old. Younger people moved out for work opportunities, and children left for education. The returning residents can be regarded as a positive sign of change in the hollow village by the implementation of the tourist program.

## 7. The Changes of the Community at the Program's the Operation Stage

This section focuses on the changes created by the program's three leisure activities, including the traditional house experience, the recreational agricultural practices and the martial arts experience.

### 7.1. The Residents' Changing Attitudes and Behaviours via the Program's Traditional House Experience

In 2019, the tourist program officially opened with 14 renovated mud-wall guest houses and 35 rooms. Both the original and the newly-built houses were owned by the residents. The company owner presumed that owning the two types of houses would make the residents feel their self-esteem enhanced; the company owner said this:

> Residents are the hosts and their guests are from the cities. In their (residents') mind, "even though they (guests) are from cities, they still have to stay at my house, while I live just besides them in my other house". Residents feel confident for themselves and treat the guests nicely.

Regarding the previous contradictory ideas between the program staff and the residents around how the traditional houses should be renovated, now the residents have changed their previous opposition to the retaining of traditional mud wall because there were more interactions among residents, program staff and guests after the official opening of the program. The evidence can be seen from a tea room (see Figure 2) transformed from an old mud-wall hut. The company owner described the increased confidence of the owner of this hut:

> Whenever I bumped into the owner of this tea room, I let him know that I only invite the most distinguished guests there. Slowly, the owner notices and feels that I really do that. Now he also feels that his hut is very important.

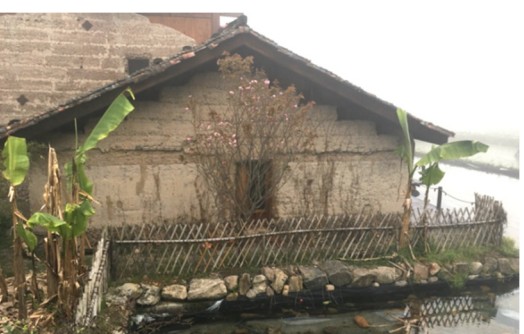 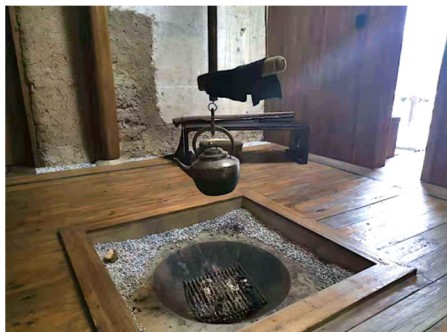

**Figure 2.** The outside and inside of the mud-wall tea room.

In addition, the program also rebuilt the village's infrastructure, including roads and walkways, drainage systems, electricity, internet connections, public toilets, and outdoor green features. Prior research has found that the higher the public infrastructure and service level, the lower the resident flight level is [64]. The consequences of gentrification shown in the prior research were that after renting their houses to tourism companies, residents must find new accommodation in adjacent towns and moved away from their homestead and the fields were left wasted [14,39]. However, these were not the cases in this tourist program.

In addition to the self-esteem boosted by the tourists' acknowledgement and improvement of physical appearance of the village, residents also gained confidence from

the new skills and the opportunities to engaged with office work other than the manual work. For instance, resident employee 2, a guesthouse room service employee, learned to use Microsoft Office Excel by observing how the managers and guesthouse receptionist conducted the administrative work with Excel. According to Manager 1, once when the receptionist was on sick leave and she could not handle all the administrative work alone, resident employee 2 used her self-taught Excel skill to share the manager' administrative workload. Now, resident employee 2 sometimes involved in the administrative work.

From the managers' point of view, the employee training provided by the program has developed the residents' skillsets and aesthetics; manager 1 said the following:

> After the guesthouse service training, our hostesses can fold a towel into the shape of a bunny and serviettes into flowers. They would go up the mountains to pick wild flowers and do DIY floral arrangements in the guest rooms. Their pursuit of beauty raises our service to another level, but it takes a while. Like before, they complaint about the wine glasses, as they're useless and easy to break while being washed. We did take some time to be able to live and work in harmony.

Manager 1 also observed this:

> The program has been running for nearly two years. We saw some of the villagers, uncles and aunties, who are service personnel for us, have changed a lot. At the beginning, they didn't understand why they should lower their voices in the restaurant and the common area. Now, sometimes, they gently remind one another, 'Shhh'.

Manager 2 shared her surprised observation of residents adopting urban tourists' practices:

> We had a couple, our guests, living here for nearly half a year. They got along well with the villagers. One day, the couple took a set of wedding photos in the village. One or two weeks later, I saw several pairs of villagers take wedding photos there too! Mr G was tilling the field in the morning, while in the afternoon, he's wearing a suit, walking alongside the field bare-footed with leather shoes in his hand, and his wife was in a white wedding dress.

### 7.2. The Environmental and Economic Changes in the Community via the Program's Recreational Agricultural Activities

The program rented a 40,002 m$^2$ unused agricultural land from the residents. The land became unused due to the long-term loss of labour, and now it was reused by planting Chinese herbal medicine, kiwi fruits and other fruit trees [62]. The two program managers developed seasonal recreational agricultural activities to attribute to the program revenues, which helped the tourist-dependent program to survive the COVID-19 pandemic. Manager 2 said the following:

> Last year we designed a 'Spring Ploughing' project, but our city members couldn't be here to do the spring ploughing, so we hired residents to help to grow corns, rapeseed flowers, rice and other stuff. When it's time to harvest, we sold and sent the cornmeal, vegetable oil, chestnuts and dry vegetables to our city members.

Earlier this year the managers developed 'the Dining Plan' project. New products, such as free-ranged chickens and eggs, spring bamboo shoots and green tea, were added into the food order list. The managers also pointed out that even though these activities aimed to increase the program revenue, the program did not over-farm or over-develop the land. With city members' recognition of the village's agricultural products, residents were now motivated to attend to their previous wasted land.

Green tea production has been the community's traditional livelihood, and it has also been developed into a leisure activity. In spring, urban members would come to pick tea leaf and help in the initial steps of green tea production. Employee nonresident 1 said that once the urban members participated in the tea harvesting and tea making processes,

they would continue to order green tea after they returned to the city. Now in spring, the program temporarily hired more residents to pick and produce the tea in order to meet the green tea orders, which provided the program and residents' additional economic gains.

The recreational agricultural activities have re-cultivated the unplanted land and restored the soil function, as well as sustained villages' traditional primarily industrial livelihood. The environmental and economic empowerment have assisted the community and residents in facing the challenge of the COVID-19 pandemic.

### 7.3. The Affirmation and Change of the Martial Arts Tradition via the Program's Martial Arts Experience

Two hundred eighty years ago, the village's martial arts were created by the residents to defend themselves from the invaders and bandits and guard the community. This set of martial arts includes three sets of boxing moves, three sets of stick-fighting moves and a set of spear-fighting moves; it had only been passed down to men in this village and had been learned by 11 generations of male residents. For this reason, practicing martial arts has been the tradition and privilege for male residents, not for women or nonresidents. The program incorporated the village's martial arts and designed it as a tourist experience, which could be seen as an affirmation to the villages' martial arts tradition. Two resident non-employees reflected on the following:

> Resident non-employee 1: My father was a martial arts master in this village. He used to say "we must pass on our martial arts and take ancestral teachings and ways of doing things seriously" . . . My father said to me I could forget about everything else except for our martial arts.

> Resident non-employee 2: My great grandfather said the same. He told us we shouldn't let our martial arts disappear in our hands. We must hand it down to the next generation.

Inspired by this tradition, the program was named as 'the hermit master'. To protect and promote this tradition, each guesthouse was named after the titles of well-known martial arts novels by Louis Cha, new plum blossom piles for training the steps were set up (Figure 3), a museum to showcase the martial arts history, techniques and masters was built (Figure 3), and the daily morning practice was organised (Figure 4).

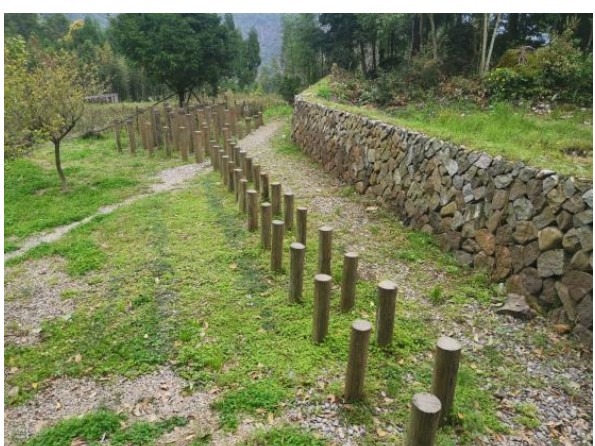 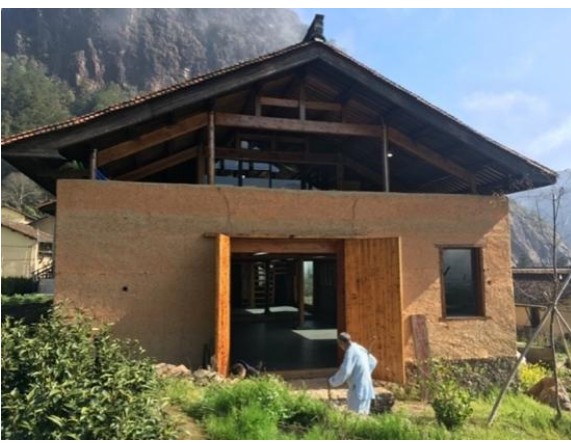

**Figure 3.** New plum blossom piles (**left**) and a newly built Martial Arts Museum (**right**).

While the program has designed activities around the martial arts, it also violated the traditional restrictions, such as not to teach women and outsiders, in order to suit the modern time. Since the operation of the program, residents have taught martial arts to tourists, regardless of their origin and gender. Both male and female martial arts teams were founded and resident employee 2 became the leader of the female team. It raised the concerns from the elders in the village; resident employee 1 said the following:

> We all learn and practice (martial arts) together. It's changed, not like before. In the past, martial arts training was very harsh and only passed to men in the village, not to women. The girls in the village weren't allowed to practice. Older folks were concerned and told us not to teach everything we know to outsiders. They were afraid that outside people would learn everything we know and the martial arts wouldn't be our own. The older folks with such concerns constantly came to us and told us what we can or can't teach to outsiders.

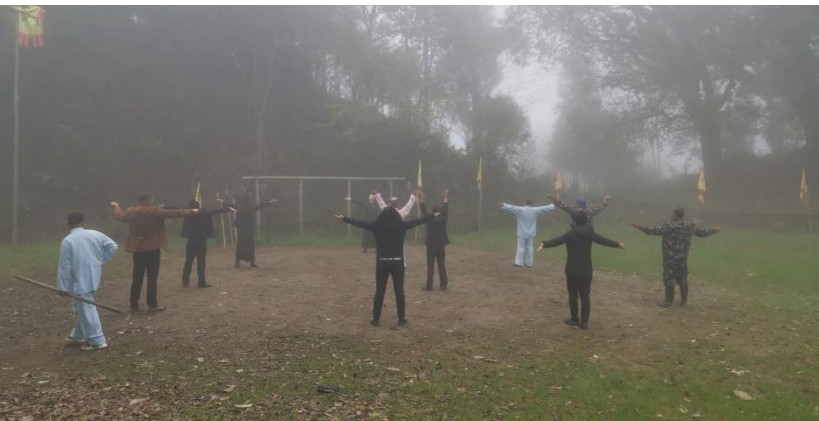

**Figure 4.** Program staff and residents' early morning martial arts practices.

As the change of the martial arts tradition, it no longer excludes female and outsiders' participation and can be learned and known by a wider group of people beyond the village residents, resident non-employee 2 affirmed the following:

> If outside people came to learn, we should teach them. Especially now, martial arts are seen as sports and can help to build up physical fitness. We need to spread our martial arts to other places, even though our ancestors made the rules not to teach women, but those are outdated ideas. If people want to be healthier and fitter through martial arts practices, I wish ours can be learned by more and more people.

In this case, the program not only promoted the village's martial arts tradition but also improved the gender equality in the community, which echoes the prior finding that in tourism development, women are likely to feel more empowered than males [65]. In the next section, we discuss what types of empowerment or disempowerment have gained by the residents from these changes in their community.

## 8. Discussion

This section discusses the varied forms of empowerment and disempowerment to the residents by the program-driven changes in the community. Because there are similar types of (dis)empowerment could be found at both the program's preparation and operation stages, we followed the conceptualisation of empowerment framework (Table 1) and categorised the (dis)empowerment into the individual- and community-levels (see Tables 3 and 4).

*At the individual level,* there are generally three types of empowerment observed, which are psychological and behavioural empowerment, economic empowerment, and knowledge and skill-set empowerment to the residents.

First, the improvement of village's infrastructure and the living conditions has contributed to residents' psychological and behavioural empowerment. The preserved traditional houses not only economically empower the residents but also can be seen as psychological empowerment to residents' self-esteem and confidence because they affirm the value of community's traditional architect style. Even though the village's living conditions may not be as modern and advanced as the urban conditions, the residents start to view their homesteads as property valuable and full of cultural meanings. In this sense, the

tourist program's protection, renovation and management of the traditional houses can be seen as the outside empowerment that transfers the residents into active agents, who explore their own interest and initiate self-transformation.

**Table 3.** Signs of empowerment and disempowerment at the individual level.

| At the Individual Level | Signs of Empowerment | Signs of Disempowerment |
|---|---|---|
| Psychological and behavioural empowerment | Residents' self-affirmation of the traditional life and living environment. | Urban residents' affirmation of the village life and the phenomena of gentrification lead to potential unequal power relationship. |
| Economic empowerment | Residents earned income from renting houses and fields and employment. | Residents were employed for lower-level job positions and their opportunity to be promoted was unclear. |
| Knowledge and skill-set empowerment | Resident-employees learned useful skillset of the tertiary industry. | Only certain knowledge and skill sets could be gained from working at the tourist program. Whether the learning could fulfil residents' own needs was unknown. |

**Table 4.** Signs of empowerment and disempowerment at the community level.

| At the Community Level | Signs of Empowerment | Questions of Disempowerment |
|---|---|---|
| Ecological empowerment | The program revived the unused and waste fields. | |
| Cultural empowerment | Promoted traditional martial arts | The traditional martial arts may become a cultural commodification and be subjected to modification. |
| Social empowerment | Houses were renovated, infrastructure was improved, resident population was increased (from about 50 to 87). | Potential unequal power relationship and social polarisation between the managerial middle-class people and the new working-class people. Returning residents were in their 40s and 50s, while no younger folks return. |
| Political empowerment | The establishment of the residential committee and residents had voices at the program's preparation and operation stages. | Resident-employees did not have the managerial roles in the program, so they may lack real influences. |

Second, the economic empowerment at the individual level was achieved by positive changes of the residents' finance. This change was made through varied ways of income offered by the program to the residents, including rents of houses and fields, employment and the sales of agricultural products.

Third, knowledge and skill-set empowerment can be seen from the evidence that resident employees learned useful skillsets of the tertiary industry through the program's training session. In particular, a resident employee acquired basic Excel skills from working at the program.

Nevertheless, from the varied changes in the village, we have also identified the phenomenon of gentrification. Unlike the overt residential displacement caused by urban gentrification, this rural tourist program did not remove original residents or cause physical displacement. However, tourism gentrification of the village did cause socio-cultural displacement, meaning that the urban middle-class needs, values and behaviours have replaced, at least partially, the residents' everyday practices and views. The relatively covert changes among residents, as well as the unequal relationship between the residents and program staff, may cause disempowerment to the residents.

Tourism gentrification can be noticed from the evidence that the program trained residents to be polite, soft-spoken and helpful staff who can provide standard hotel and restaurant services to idyllic-lifestyle seeking tourists. Residents' pervious behaviours were deemed as 'bad' or 'uneducated', and hence needed to be changed via training. The combination of urban practices and rural environment could meet the middle-class tourists' needs and the programs' business interests. The program created an oasis of tertiary industry within the area predominated by the primary industry. The skillsets useful in the tertiary industry and urban aesthetics were not a part of residents' traditional life or even useful outside the program (e.g., in the agricultural industry), but they were highly regarded by the managers as a positive change in the community. Resident employees' behavioural changes implied the external gentrification and internal self-gentrification. However, due to their limited skills, the residents were situated at the lowest level of job positions in tertiary industry, such as cleaners, waitresses, handymen and gardeners.

The unequal relationship between the residents and program staff would produce potential disempowerment to the residents and uncertainties to the meaningful transformation of the community. For instance, the residents were employed as lower-level job positions, and the opportunity to be promoted was unclear. Residents' social–economic status is much lower than the program managers and company owner. During the focus group and casual conversations with the company owner and the program managers, we notice that they consciously and consistently referred to the residents as employees or hosts/hostesses, which reflects a new working identity attached to the residents. In addition, regarding the knowledge empowerment, only the given set of knowledge and skills could be gained from working for the tourist program. Whether the learning outcome could fulfill residents' own needs is unknown.

*At the collective level*, there are mainly four types of empowerment observed, which are ecological, cultural, social and political empowerment to the community incurred by the changes of the community.

First, the ecological empowerment came from the revival of the unused fields through the recreational agricultural activities and the sales of organic products. The environmental change not only empowered the community, but also brought the economic improvement at the individual level.

Second, cultural empowerment occurred as the program has respected, protected and promoted the community's 280-year-long practices of martial arts. The changing martial arts traditions especially empowered the female residents, making them feel more involved in the community and the program.

Third, the strong signs of social empowerment were the returning residents and the increasing population since the program's preparation stage.

Fourth, political empowerment was gained through the establishment of the residential committee. It has provided the residents a platform to voice their opinions at the program's preparation and operation stages. Although it was largely a top-down tourism development project by a company, the residential committee provided ways for residents to become involved. It differs from the issue raised from the prior research on community-based tourism in China that governments and companies often imposed or single-handily decided the tourism development [29,30].

Despite the community-level empowerment resulting from the changes, we also found the signs of social and political disempowerment produced by the tourism gentrification. For example, the traditional martial arts were treated as a cultural commodification and were subjected to modification. Returning residents were all in their 40s and 50s, while no younger residents return. These aspects may lead to social disharmony and unstainable change in the population. Political empowerment may be compromised by the resident employees' lower-level roles in the program, which may decrease their real influence on the decision making at the managerial level.

Based on the empowerment and disempowerment to the residents and community, it is believed that this tourist program has positively changed (gentrified) the hollow village but also produced new uncertainties and questions at the same time.

## 9. Conclusions

This research explores the multifaceted changes in the community induced by the implementation of the tourist program and the multiple forms of (dis)empowerment to the residents. From the forms of (dis)empowerment at the individual and the community level, we discussed to what extent a hollow village could be meaningfully and sustainably transformed through a community-based tourist program. Findings of this small single case study largely demonstrated the positive transformations have taken place in the community and among individuals. First, at the program preparation stage, different participatory choices for residents were provided by this outsider-initiated tourist program. Hence, a relatively equal dialogue between the community and the program has laid a foundation for the meaningful transformation. Second, findings demonstrate that the program activities have empowered the residents at economic, psychological and knowledge levels, and the community at social, cultural and ecological levels. These multiple empowerment-driven transformations have met residents' diverse needs. In particular, from a resident's self-taught IT skills, we see the shift from outside-empowerment to self-empowerment. Third, it is worth noting that under the COVID-19 pandemic, with fewer tourists, the program still generates revenue and retains employees by selling and delivering the agricultural products to urban members. This evidence demonstrates the sustainable capability of this program and the benefits for the residents and community.

Nevertheless, this study also points out the disempowerment to residents and community caused by tourism gentrification and the unequal relationship between the residents and program managerial staff. Residents' behavioural, knowledge and economic disempowerment could be caused by the uncritical acclamation of the urban lifestyle, the limited learning opportunities, and lower job positions offered by the program. As to social and political disempowerment, it is found in the community's unsustainable change of the population and the lack of real influences to the decision making at the managerial level. These concerns of the disempowerment may indicate the future research directions.

**Author Contributions:** Conceptualization, Z.Z., Y.W. and L.L.; literature review, Y.W. and Y.O.; methodology, L.L., Y.O. and Y.W.; formal analysis, L.L., Y.W., Y.O. and Z.Z.; writing—original draft preparation, Z.Z., Y.O. and Y.W.; writing—review and editing, L.L., Y.W., Y.O. and Z.Z.; tables and images, Y.W. and Y.O.; supervision, L.L.; project administration, Z.Z.; funding acquisition, L.L. All authors have read and agreed to the published version of the manuscript.

**Funding:** This research was funded by The National Social Science Fund of China, grant number 19FTYB005.

**Institutional Review Board Statement:** The study was approved by the Ethics Committee at the Department of Psychology and Behavioural Science of Zhejiang University, NO. 2021[072].

**Informed Consent Statement:** Informed consent was obtained from all subjects involved in the study.

**Data Availability Statement:** No new data were created or analysed in this study. Data sharing is not applicable to this article.

**Acknowledgments:** The authors would like to thank the tourist program staff and the villagers.

**Conflicts of Interest:** The authors declare no conflict of interest.

## Appendix A

Focus group Interview guide

1. Could you tell us about your role in this program or in the village? Could you tell us about how long have you been working in this program or living in the village?

2. Do you know why the company chose to develop the tourism program in this village? What were the resources in the village that attracted the company?
3. Could you tell us about the interaction between the company/program managers and the residents? Were there anything interesting or unforgettable?
4. Since the program implementation, what changes have happened in the village that you could recall? Do you like these changes, why?
5. In your opinion, how this program has benefited the community?
6. Was there any difficulty in the program implementation? How did you solve the problem?
7. Could you tell us about the village's traditional martial arts?
8. Generally, do you satisfy with the situation now in the village, in what ways?
9. Anything you would like to tell us about the program or the village or the community?

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
