# Peer review of "Between Empowerment and Gentrification: A Case Study of Community-Based Tourist Program in Suichang County, China"

_sustainability, doi:10.3390/su14095187_

Round 1
Reviewer 1 Report
Looking at the manuscript proposed for publication, I believe that it would benefit from:
- refining the structure and changing the positioning of sections 3 and 4, changing their places, in view of the logic set out in the abstract;
- in line 94 and the next paragraph would be good to edit the text: there is information from researches on tourism and its effects presented as direct observation, while in fact it represents researches by other authors which means that the citation needs to be clarified;
- Figure 1 lacks the source - in general it is good to review the illustrative materials and to specify the sources and their recording;
- on line 188 there is a repetition of words and so the meaning of what is said is lost;
- in lines 190-193 the text also needs editing - how many programs are there: one or more than one, it is not clear is the mentioned tourist company working in other places in China, ect.?;
- In terms of methodology, there is the issue of the two focus groups under consideration - however, one of them is smaller and goes beyond the theoretical framework for structuring a focus group. Perhaps it would be good for the authors to think about this moment and its presentation - it is about the group of representatives of the tourist company;
- there are some stylistic errors, especially in connection with the used verb tense to expose the author's thoughts;
- line 319-322 - the paragraph must be edited stylistically and grammatically;
- table 2 also has no source too. It concludes section 8 and immediately follows the Conclusion section. It would be good to have a summary comment after the table, as it is important in the context of the results and the discussion. In fact, section 8 has the character of a discussion, but a real one is not set as a compositional element of the article. This finding is the reason for my recommendation.
Author Response
Cover letter
We appreciated the reviewers’ constructive comments and revised the manuscript accordingly. Since the manuscript has been substantially revised, some comments may not be applicable to this revision. Our responses are listed below.
Reviewer 1
- refining the structure and changing the positioning of sections 3 and 4, changing their places, in view of the logic set out in the abstract;
Response: We have restructured the article and rewritten the abstract and the main body.
- Figure 1 lacks the source - in general it is good to review the illustrative materials and to specify the sources and their recording;
Response: Figure 1 was based on an online map (the source has been credited in the writing). We especially dotted the location of the village on the map.
- on line 188 there is a repetition of words and so the meaning of what is said is lost;
Response: The sentence has been rewritten.
- In terms of methodology, there is the issue of the two focus groups under consideration - however, one of them is smaller and goes beyond the theoretical framework for structuring a focus group. Perhaps it would be good for the authors to think about this moment and its presentation - it is about the group of representatives of the tourist company;
Response: We admitted that the demographics of the participants cannot represent all the local residents. We aimed to invited the participants who had abundant experiences of working in the program as well as living in the village. Hence, their experiences are helpful for us to address the research questions. Most of the elder resident-nonemployees spoke the local dialect, which the researchers were unable to speak. This is a limitation in data collection and we have pointed out in the writing.
- there are some stylistic errors, especially in connection with the used verb tense to expose the author's thoughts;
Response: We have edited the grammar and has proofread this draft.
- line 319-322 - the paragraph must be edited stylistically and grammatically;
Response: this paragraph has be revised.
- It concludes section 8 and immediately follows the Conclusion section. It would be good to have a summary comment after the table, as it is important in the context of the results and the discussion. In fact, section 8 has the character of a discussion, but a real one is not set as a compositional element of the article.
Response: now the section 8 has been reorganised, and the discussion has been added.
Reviewer 2 Report
The research deals with a tiny case study in China but is somewhat informative. The introduction is a good synthesis of current knowledge, even if the authors should explicitly define community-based tourism. I made some minor change suggestions in the text. The paper will gain to be edited regarding language, especially tense uses.

Author Response
We appreciated the reviewers’ constructive comments and revised the manuscript accordingly. Since
Reviewer 2
The research deals with a tiny case study in China but is somewhat informative. The introduction is a good synthesis of current knowledge, even if the authors should explicitly define community-based tourism. I made some minor change suggestions in the text. The paper will gain to be edited regarding language, especially tense uses.
Response: A definition for “community-based tourism” has been provided in the second paragraph, section 1 introduction.
We also make changes according to the reviewer 2’s PDF. We used the simple past tense and past perfect tense.
Reviewer 3 Report
I read with interest the manuscript entitled 'Transforming a Chinese hollow village: The potentials and uncertainties of community-based tourist program'. I strongly encourage the authors to rethink the research objectives and hypotheses they may define based on the material collected.
Key issues of concern:
1. Lack of consistency in detailing of the issues presented. After a very general introduction, without analysis of the context and specificity of the case study, conclusions are drawn on the basis of fragments of the interviewees' statements.
2. It is not explained how the surveyed group is representative for the local community (7 people employed in the programme, including the owner, and only 2 residents not employed in the programme).
3. Key informations, among which demographic or ownership structure, are scattered throughout the text and often contradictory (cf. lines 314-315 and 544).
4. Discussion which is a required element (cf. Guidelines for Authors & Template) is lacking.
5. The manuscript needs extensive English editing. Reading is hampered by numerous syntax, spelling and punctuation errors, basic terms (including gentrification and empowerment) are used inconsistently.
6. The authors avoid the difficult question regarding the role of a top-down approach in limiting both the empowerment mechanism and the bottom-up / community-based character of the programme. The same goes for the sustainability of the whole process (renovation of old houses for a closed group of users registered in the programme versus building new concrete houses, decorated to look like old ones for the residents).
Author Response
We appreciated the reviewers’ constructive comments and revised the manuscript accordingly. Since the manuscript has been substantially revised, some comments may not be applicable to this revision. Our responses are listed below.
Reviewer 3
- Lack of consistency in detailing of the issues presented. After a very general introduction, without analysis of the context and specificity of the case study, conclusions are drawn on the basis of fragments of the interviewees’ statements.
Response: The manuscript has been substantially rewritten to improve the consistency. As to the context of the case, more background information of the village has been added.
- It is not explained how the surveyed group is representative of the local community (7 people employed in the programme, including the owner, and only 2 residents not employed in the programme).
Response: We admitted that the demographics of the participants cannot represent all the local residents. We aimed to invite the participants who had abundant experiences of working in the program as well as living in the village. Hence, their experiences are helpful for us to address the research questions.
- Key information, among which demographic or ownership structure, are scattered throughout the text and often contradictory.
Response: In this version, the information about this program has been introduced in two places: ‘1. Introduction’ and ‘4. Tea Village and the Tourist Program’. In ‘1. Introduction’, the information helped to explain why this program has been chosen for the study. Then, more detailed information about the program was presented in ‘4. Tea Village and the Tourist Program’.
- Discussion which is a required element (cf. Guidelines for Authors & Template) is lacking.
Response: The section of the discussion has been added.
- The manuscript needs extensive English editing. Reading is hampered by numerous syntax, spelling and punctuation errors, basic terms (including gentrification and empowerment) are used inconsistently.
Response: An English native speaker has proofread this draft.
- The authors avoid the difficult question regarding the role of a top-down approach in limiting both the empowerment mechanism and the bottom-up / community-based character of the programme. The same goes for the sustainability of the whole process (renovation of old houses for a closed group of users registered in the programme versus building new concrete houses, decorated to look like old ones for the residents).
Response: In the discussion section, we discuss the changes that could be considered as empowerment or disempowerment. From empowerment or disempowerment, we then discuss what types of changes were meaningful for the village.
Round 2
Reviewer 3 Report
Dear Authors,
My compliments on the revised version of your manuscript. I appreciate your efforts and hard work. Allow me to point out some minor inconsistencies:
- Line 62 ends with the term "self-gentrification". Consider moving the references made in line 114 [37,45] here;
- Start all quotations with capital letters (cf. lines 298 and 375 vs lines 326, 340, 363, etc.);
- Consider adding an Appendix with all the interviews;
- Last but not least, since your paper is a case study rather than in-depth research on the mechanisms of gentrification or empowerment, do not hesitate to inform the reader. How about a simple, informative solution: "Between empowerment and gentrification: Community-based tourist program in Suichang County, China case study" (or any variant that will not make too big a promise)?
I wish you all the luck and successful progress in your research.
Author Response
We appreciated the reviewer’s thorough feedbacks. The changes have been made in accordance.
- Line 62 ends with the term "self-gentrification". Consider moving the references made in line 114 [37,45] here;
Response: We have made the change according your advices.(cf. line 62 and 114)
- Start all quotations with capital letters (cf. lines 298 and 375 vs lines 326, 340, 363, etc.)
Response: We have used capital letters to started the quotations.
- Consider adding an Appendix with all the interviews.
Response: We have added the ‘Appendix-Focus group Interview guide’ at the end of the article.
- Last but not least, since your paper is a case study rather than in-depth research on the mechanisms of gentrification or empowerment, do not hesitate to inform the reader. How about a simple, informative solution: "Between empowerment and gentrification: Community-based tourist program in Suichang County, China case study" (or any variant that will not make too big a promise)?
Response: Thanks for your suggestion. We followed your advice and changed the tittle to "Between empowerment and gentrification: A case study of community-based tourist program in Suichang County, China".